# Mealtime Regularity Is Associated with Dietary Balance among Preschool Children in Japan—A Study of Lifestyle Changes during the COVID-19 Pandemic

**DOI:** 10.3390/nu14142979

**Published:** 2022-07-21

**Authors:** Yuki Tada, Yukari Ueda, Kemal Sasaki, Shiro Sugiura, Mieko Suzuki, Hiromi Funayama, Yuka Akiyama, Mayu Haraikawa, Kumi Eto

**Affiliations:** 1Department of Nutritional Science, Faculty of Applied Bioscience, Tokyo University of Agriculture, Sakuragaoka 1-1-1, Setagaya 156-8502, Tokyo, Japan; 2Department of Health and Nutrition, Faculty of Health and Nutrition, Osaka Shoin Women’s University, Hishiyanishi 4-2-26, Higashiosaka 577-8550, Osaka, Japan; ueda.yukari@osaka-shoin.ac.jp; 3Department of Food and Health Sciences, Jissen Women’s University, Osakaue 4-1-1, Hino 191-8510, Tokyo, Japan; kemal-s@umin.org; 4Maternal and Child Health Center, Aichi Children’s Health and Medical Center, Morioka 7-426, Obu 474-8710, Aichi, Japan; bee_jayz@hotmail.com; 5Department of Early Childhood Care and Development, Tamagawa University, Tamagawagakuen 6-1-1, Machida 194-8610, Tokyo, Japan; mieko@edu.tamagawa.ac.jp; 6Department of Pediatric Dentistry, Tsurumi University School of Dental Medicine, Tsurumi 2-1-3, Tsurumi, Yokohama 230-8501, Kanagawa, Japan; funayama-h@tsurumi-u.ac.jp; 7Department of Health Sciences, University of Yamanashi, Shimokato 1110, Chuo 409-3898, Yamanashi, Japan; yukaa@yamanashi.ac.jp; 8Department of Child Studies, Seitoku University, Iwase 550, Matsudo 271-8555, Chiba, Japan; mharai@wa.seitoku.ac.jp; 9School of Nutrition Sciences, Kagawa Nutrition University, Chiyoda 3-9-21, Sakado 350-0288, Saitama, Japan; ekumi@eiyo.ac.jp

**Keywords:** COVID-19, dietary balance, Japan, mealtime regularity, preschool children

## Abstract

The novel coronavirus-19 (COVID-19) pandemic has considerably impacted children’s lives. The aim of this study was to determine whether the pandemic affected mealtime regularity among preschool children and whether maintaining regular mealtimes or changes in mealtime regularity during the pandemic were related to dietary balance, including chronological relationships. This online cross-sectional survey involving individuals registered with a company that provides meals to children aged 2−6 years was conducted in February 2021. Using a 40-point scale, a healthy diet score (HDS) was developed to evaluate children’s dietary balance. The participants were divided into four groups based on their responses, and multiple regression analyses were performed with the HDS as the dependent variable. Maintaining regular mealtimes was associated with practices such as waking and going to bed earlier, less snacking, and eating breakfast every day. Even after adjusting for basic attributes, lifestyle habits, household circumstances, and other factors, regular mealtimes were still positively correlated with the HDS. These findings indicate that maintaining regular mealtimes is associated with higher HDS scores and better lifestyle habits. Furthermore, as the changed HDS was higher in the group whose mealtimes became regular during the pandemic, adopting regular mealtimes may lead to a more balanced diet.

## 1. Introduction

Since the novel coronavirus-19 (COVID-19) pandemic began in 2020, it has had a considerable impact on children’s lives. In Japan, the first state of emergency was declared on 7 April 2020, which was accompanied by long-term closures of kindergartens, elementary schools, middle schools, and high schools. Many schools resumed in June but with special measures in place, such as distributed attendance and shortened classes. Since then, children have experienced a prolonged period of stress from changes, such as shorter summer vacations, fewer or cancelled events, and restricted club activities. As of June 2022, children still have to follow this “new normal” in various aspects of life.

It has been reported that worldwide, how and what children eat has changed during the pandemic. In a previous study in Italy among children aged 6−18 years with obesity, there was a significant increase in snack and soft drink consumption following the implementation of the COVID-19 lockdown [1]. In a survey of families with one or more children aged 18 months to 5 years in Canada, many respondents indicated that the COVID-19 pandemic had increased their consumption of snack foods [2]. Additionally, a study conducted by Androutsos et al. among children aged 2−18 years in Greece revealed that during the COVID-19 lockdowns, weight gain was associated with higher intakes of pasta, sweets, and snacks, as well as reduced physical activity [3]. In a survey of students in elementary school and older children in Japan, many indicated that both the opportunities for and the quantity of snacks had increased compared with before COVID-19 [4]. This shows how the spread of the novel coronavirus has had a considerable impact on children’s lifestyle habits and diets.

However, in other studies only some children were found to have deteriorated lifestyle and eating habits during the pandemic. In the study by Adroutsos et al., no changes were observed with respect to aspects such as sleep and the frequency of eating unhealthy foods, such as fast foods, sweets, and snacks, in a majority of the children surveyed [3]. Furthermore, in a study investigating the factors associated with healthier or unhealthier eating habits in adults during the pandemic, 71.6% of the participants did not change their eating habits [5]. Among the factors positively associated with healthier dietary habits were annual household income, changes in household income, COVID-19 morbidity among friends, health literacy, and exercise frequency [5]. A change in caregivers owing to COVID-19 is thought to affect the eating habits of children living in the same household. 

It is easier for children to maintain set wakeup times and mealtimes when the time at which they leave for and arrive home from school are constant. Therefore, the closure of educational institutions and child welfare facilities made it difficult to maintain set wakeup and bed times, which may have further affected children’s eating habits. Following a survey of students in elementary schools and older children in Japan, it was reported that compared with before COVID-19, 30% of students in the fourth grade and above in elementary school had later or disturbed wakeup/bed times, and although indicated for less than 10% of the children, there was also an increase in irregular mealtimes and skipped meals [5]. In the 2015 National Nutrition Survey on Preschool Children in Japan, there a higher percentage of children indicated that they always eat breakfast among those with earlier wake-up and bed times [6]. It was further reported that later and more irregular meal and sleep times were associated with poorer physical and mental states in infants [7]. Previous studies among adults reported that mealtimes were associated with the diurnal rhythm of cardiac autonomic activity [8,9], highlighting the importance of maintaining regular mealtimes from early childhood. However, to the best of our knowledge, there have been few reports on whether regular mealtimes are associated with lifestyle habits and dietary balance in preschool children. In particular, it is unclear whether the dietary habits of children who maintained their mealtime regularity during the pandemic are more balanced than those who did not.

Therefore, with this study, we aimed to determine whether the pandemic affected mealtime regularity among preschool children and to compare children who maintained regular mealtimes during the pandemic with those who did not in order to determine the relationship with their lifestyle habits, including wakeup and bed times, as well as dietary balance. Furthermore, we aimed to investigate whether pandemic-related changes in mealtime regularity were related to changes in dietary balance, including chronological relationships.

## 2. Materials and Methods

### 2.1. Study Participants and Design

This online cross-sectional survey involving individuals registered with Cross Marketing Group (C company, Shinjuku, Tokyo), a company that provides meals to children aged 2−6 years, was conducted during the period of 24−25 February 2021. First, a screening survey was performed on all individuals registered with C company. Those who indicated that they had children aged 2−6 years and were the meal provider for the oldest child in this age range were eligible to participate in the study. Thereafter, the participants were sorted by region at ratios close to the distribution of households with children aged <6 years in the 2015 national census, and 2000 respondents were recruited until those figures were reached. All child-related questions pertained to the oldest child among those aged 2−6 years.

The questions included the respondent’s age, residential prefecture, employment status, child’s age, child’s sex, birth order, primary daytime daycare location, and health status. The participants were also asked about current eating and lifestyle habits and whether these had changed due to the COVID-19 pandemic. These questions were designed from a maternal and child health perspective and were based on those used in the 2015 National Nutrition Survey on Preschool Children, Japan (Ministry of Health, Labor, and Welfare of Japan) [6].

### 2.2. Calculating the Healthy Diet Score (HDS)

A healthy diet score (HDS) was developed to comprehensively evaluate children’s dietary balance. One goal of Japan’s Fourth Basic Plan for the Promotion of Shokuiku is to increase the frequency of meals that include a combination of staple, main, and side dishes [10]. The Japanese Food Guide Spinning Top, which was created by the Ministry of Agriculture, Forestry, and Fisheries and the Ministry of Health, Labor, and Welfare of Japan, uses five categories to determine diet quality as follows: grain dishes, vegetable dishes, fish and meat dishes, fruit, and milk (dairy products) [11]. These five categories were used when categorizing meals in this study. In addition, unhealthy foods that should be avoided, such as fast foods and sweets, were evaluated as reverse-scored questions. Specifically, of the 13 food groups included in the food frequency survey, grains; fish; meat; eggs; soybeans/soy products; vegetables; fruit; milk/dairy products; and unsweetened beverages, such as tea were categorized as healthy, whereas sweetened beverages, sweets, instant noodles/cup ramen, and fast foods were categorized as unhealthy. The intake frequency of foods from the healthy food groups was scored as follows (Figure 1). Grains and vegetables: ≥2 times per day = 4 points, 1 time per day = 3 points, 4−6 times per week = 2 points, ≤3 times per week = 1 point, and <1 time per week = 0 points; and fish, meat, eggs, soybeans/soy products, fruit, milk/dairy products, and unsweetened beverages, such as tea: ≥1 time per day = 4 points, 4−6 times per week = 3 points, 1−3 times per week = 2 points, and <1 time per week = 1 point. The intake frequency of unhealthy foods, such as sweetened beverages, sweets, instant noodles/cup ramen, and fast foods, was reverse-scored as follows: ≥2 times per day = 0 points, 1 time per day = 1 point, 4−6 times per week = 2 points, 1−3 times per week = 3 points, and <1 time per week = 4 points. A mean value was calculated using the scores of all the main dishes (fish, meat, eggs, and soybeans/soy products). The scores for grain dishes, main dishes (mean); side dishes; fruit; dairy products; unsweetened beverages, such as tea; and unhealthy foods (reverse-scored) was totaled to obtain the HDS (0−40 points). The following secondary indicators were calculated based on intakes of: (1) grain dishes, main dishes (mean), and side dishes (0−12 points); (2) grain dishes, main dishes (mean), side dishes, fruit, and dairy products (0−20 points); (3) grain dishes, main dishes (using the total instead of the mean to consider the variety of main dishes), and side dishes (0−32 points); and (4) infrequently consuming unhealthy foods (0−16 points).

To evaluate the validity of the HDS indicator developed for this study, the intake frequency of each food group was determined using the food diversity score (FDS) developed by Ishikawa et al., which is calculated using the same questions [12]. FDS is an 8-point scale that evaluates dietary diversity (grains, fish, meat, eggs, soybeans/soy products, vegetables, fruit, and milk/dairy products) based on the Food and Agriculture Organization’s dietary assessment [13]. The correlation coefficient between the FDS and HDS was 0.574 (*p* < 0.001), indicating a significant positive correlation. A comparison of the FDS among the four groups based on mealtime regularity yielded similar results to comparisons using the HDS (highest score was in the group that originally had regular mealtimes that did not change). However, the differences between the four groups were larger using the HDS than the FDS. This indicates that the HDS is a useful indicator that considers the diversity of the food groups recommended in the Japanese diet (grain, main, and side dishes), as well as the infrequent consumption of unhealthy foods, such as sweets and instant noodles. 

### 2.3. Changes in the HDS during COVID-19: Changed-Healthy Diet Score (C-HDS)

Here, we aimed to determine whether the quality of meals improved or deteriorated as mealtimes became more regular or irregular, respectively. The changed-healthy diet score (C-HDS) was calculated based on changes in the intake frequency of each food group compared with before COVID-19. The food categories used were the same as those for the HDS. Healthy foods included grains (rice or bread, for example); fish; meat; eggs; soybeans/soy products; vegetables; fruit; milk/dairy products; and unsweetened beverages, such as tea. Unhealthy foods included sweetened beverages, sweets, instant noodles/cup ramen, and fast foods. The change in intake frequency for a food group before and after COVID-19 was scored as follows: for increased intakes of a specific food group, healthy foods = +1 point, unhealthy foods = −1 point, and no change = 0 points (increased score: −4 to 9 points); and for decreased intakes of a specific food group, healthy foods = −1 point, unhealthy foods = +1 point, and no change = 0 points (decreased score: −9 to 4 points). These scores were totaled to determine the C-HDS. However, as 69.9% of the participants had a score of 0 points (no change), the scores were grouped into three categories for the regression analysis as follows: increased, no change, and decreased. 

### 2.4. Ethical Considerations

After viewing the guidelines and principles of the Japan Marketing Research Association as summarized by company C and an explanation of the survey, informed consent to participate in the online survey was obtained. The survey explanation informed participants that the collected data would undergo statistical processing and be published in a manner whereby the individuals could not be identified and that the data would not be used for purposes other than the survey. Ethical approval was obtained from the Human Research Ethics Committee of Kagawa Nutrition University (approval number 317). 

In addition, the survey data were used in a secondary analysis as part of an administrative promotion project under a Japanese Ministry of Health, Labor, and Welfare grant, “Research for Effective Development of Nutrition and Dietary Support for Healthy Development in Early Childhood” (grant number: 20DA2002). 

### 2.5. Analysis

#### 2.5.1. Data Set for Analysis

To prevent impersonation and fraudulent responses, 1 participant who did not answer almost any of the questions and 17 participants who did not provide answers about their relationship to the child, the child’s sex, or birth order were excluded from the analysis. Furthermore, 2 participants who were grandparents, 29 participants who did not provide answers for questions addressing changes in meal regularity, and 101 participants who indicated that they did not want to provide answers about their dietary intake frequency at present or before/after COVID-19 were excluded from the analysis; a total of 1850 participants were included in the final analysis (Figure 2).

Inconsistent answers for individual items were excluded from the final analysis. For example, answers that indicated that the respondents’ children are now “rarely” involved in cooking but “considerably more” than before the COVID-19 (*n* = 2) were considered to be inconsistent answers. Regarding a child’s height and weight, values above the 99th percentile or below the 1st percentile and calculated using the least squares mean method were considered input errors and excluded from the calculation of obesity (height, *n* = 71 participants; weight, *n* = 33 participants).

#### 2.5.2. Statistical Analysis

The participants were divided into four groups based on their responses to questions addressing mealtime regularity after COVID-19 as follows: became regular (*n* = 125), originally regular and unchanged (*n* = 1514), became irregular (*n* = 63), and originally irregular and unchanged (*n* = 148). The distribution of independent variables was examined using the χ^2^ test (residual analysis when a significant difference was noted) or one-way analysis of variance (multiple comparisons using the Games−Howell method when a significant difference was noted). 

To determine the relationship between the current HDS and mealtime regularity, simple regression analyses were performed with the HDS as the dependent variable and the respondents’ attributes, child’s attributes, lifestyle habits, and household circumstances as independent variables. In model 1, multiple regression analysis was performed with the HDS as the dependent variable for each classification of the independent variables. In model 2, multiple regression analysis was performed with forced entry method of the items that exhibited significant associations with the HDS in model 1, the relationship of the respondent, mother’s employment situation, child’s sex, child’s age, BMI percentile, and mealtime regularity. Mealtime regularity was converted into a binary dummy variable (1, originally regular; 0 otherwise). To account for the influence of missing values (BMI percentile (*n* = 155), others), the multiple regression analysis used for model 2 was performed with multiple imputation of missing values. 

To examine the relationship between changes in the HDS (C-HDS) and in mealtime regularity after COVID-19, multiple logistic regression analysis was performed with the C-HDS as the dependent variable and the respondent’s attributes, child’s attributes, pandemic-related changes in the child’s lifestyle and diet, and household circumstances as independent variables. A C-HDS score of ≤0 points was treated as a decrease, and a C-HDS score of ≥1 point was considered an increase. A C-HDS score of 0 points (no change) was used as the reference value in the multiple logistic regression analysis. A multiple logistic regression analysis was performed with forced entry of the following independent variables: changes in lifestyle habits and household circumstances that were significantly associated with the current HDS and mealtime regularity, respondent’s relationship, mother’s employment status, child’s sex, child’s age, and BMI percentile. Mealtime regularity was converted into a 3-value variable (1, became irregular; 2, became regular; 3, no change). Multiple logistic regression analysis was performed by multiple imputation of missing values. The Statistical Package for the Social Sciences, version 28 (IBM SPSS Inc., Chicago, IL, USA), was used for all statistical analyses, and statistical significance was set at *p* < 0.05.

## 3. Results

### 3.1. Comparison of Participant Characteristics according to Changes in Mealtime Regularity Compared with before COVID-19

Table 1 shows the participant characteristics according to changes in mealtime regularity compared with before COVID-19. In 41.6% of participants, the relationship to the child was through the father and in 58.4%, through the mother. 

Table 2 shows the lifestyle habits and household circumstances according to the change in mealtime regularity compared with before COVID-19. There were significantly more participants in the originally regular group who woke up before 8 a.m. and who went to bed before 10 p.m. on weekdays and holidays. Screen time on weekdays was <2 h in 79.3% of participants, and this percentage was significantly higher in the originally regular group. Screen time on holidays was ≥2 h in 35.7% of participants, and this percentage was significantly higher in the originally irregular group. Significantly fewer participants in the ‘became irregular’ group defected almost every day. Regarding dietary status, significantly more participants in the originally regular group indicated that they snacked 0−1 time per day and ate breakfast every day. 

Overall, 81.6% and 96.1% of the respondents indicated that their child ate breakfast and dinner with an adult, respectively. Significantly more children ate breakfast with an adult in the originally regular group, and significantly fewer ate dinner with an adult in the originally irregular group. Regarding the frequency of cooking, there were no significant differences between the groups. Significantly more participants in the originally irregular and unchanged groups indicated that they lacked financial security, and significantly more participants in the ‘became irregular’ group indicated that they had little free time. 

### 3.2. Comparison of the Healthy Diet Score (HDS) according to Changes in Mealtime Regularity Compared with before COVID-19

Table 3 shows the HDS according to changes in mealtime regularity. The current HDS (0−40 points) of the originally regular and unchanged group was 31.6 ± 4.0 points, which was significantly higher than in the other groups. Intakes of grain dishes, main dishes (mean), and side dishes (0−12 points); grain dishes, main dishes (mean), side dishes, fruit, and dairy products (0−20 points); and grain dishes, main dishes (total), side dishes, fruit, and dairy products (0−32 points) were significantly higher in the ‘became regular’ group and originally regular and unchanged groups compared with the originally irregular and unchanged group. Significantly more participants infrequently consumed unhealthy food (0−16 points) in the originally regular group than in the ‘became regular’, originally irregular, and unchanged groups.

Regarding changes in food group intakes before and after COVID-19, the increased score (−4 to 9 points) was significantly higher in the ‘became regular’ group than in the originally regular and unchanged group, as well as the originally irregular and unchanged group. The decreased score (−9 to 4 points) was significantly lower in the ‘became irregular’ group than in the originally regular and unchanged group, as well as the originally irregular and unchanged group. The mean C-HDS (−13 to +13) in the ‘became regular’ group was 0.8 ± 2.3 points, which was significantly higher than in the other groups. 

### 3.3. Multiple Regression Analyses with the Healthy Diet Score (HDS) as the Dependent Variable

Table 4 shows the results of the multiple regression analyses of the factors associated with the HDS. The HDS was significantly higher in the originally regular group and, even when adjusted for the other factors, was significantly associated with the investigated basic attributes of wakeup time (weekdays and holidays), bed time (weekdays and holidays), physical activity time (holidays), defecation frequency, snack frequency, frequency of breakfast for the child, eating together (breakfast and dinner), cooking frequency, and financial resources (β = 0.131, *p* < 0.001).

### 3.4. Multiple Logistic Regression Analysis with the Change in Healthy Diet Score (C-HDS) as the Dependent Variable

Table 5 shows the results of the multiple logistic regression analysis using the C-HDS as the dependent variable. A significant increase in C-HDS was noted in the ‘became regular’ group (odds ratio: 2.21, 95% confidence interval: 1.35−3.61). A decreasing trend was observed in the C-HDS in the ‘became irregular’ group. 

## 4. Discussion

In addition to investigating how children’s mealtime regularity relates to their current lifestyle habits and dietary status, we also investigated whether pandemic-related changes in mealtime regularity were associated with changes in dietary balance, including chronological relationships. Here, the results showed that maintaining regular mealtimes was associated with practices such as waking and going to bed earlier, less snacking, and eating breakfast every day. Furthermore, even after adjusting for basic attributes, lifestyle habits, household circumstances, and other factors, regular mealtimes were still significantly positively correlated with the HDS, whereas a change in regular mealtimes was significantly associated with an increased C-HDS. 

Maintaining regular mealtimes has previously been shown to be associated with positive lifestyle habits. A cluster analysis using meal and sleep time data in preschool children after forming clusters, such as a group in which both meal and bed times were extremely late, a group with late and irregular times, and a group that woke early and regularly, showed that meal and sleep times tended to be linked [7]. In a previous study, missing breakfast was associated with negative lifestyle habits in primary school children, such as increased intake of soft drinks, high screen time, and low physical activity levels [14]. The “Health guide starting with diet—Raising children who enjoy eating” of the Japanese Ministry of Health, Labor, and Welfare describes the importance of “feeling hungry and having an appetite, then experiencing the pleasure of properly satisfying this” as a target for healthy child development through food and states that it is ideal for children to have a dietary rhythm [15]. Here, our findings were consistent with previous research, in that more participants in the group that originally had regular mealtimes indicated that they eat breakfast every day and wake up and go to bed early. In addition, even after adjusting for lifestyle habits and other factors, originally having regular mealtimes was positively associated with the HDS. 

Having regular mealtimes means avoiding missing meals and maintaining constant intervals between meals. Moreover, in a study of children aged 7−18 years, missing breakfast, lunch, or dinner was associated with low vegetable and fruit intakes [16]. High intake of junk food have also been reported to increase the risk of missing meals [17]. Furthermore, a study of children aged 12−16 years reported that an irregular diet and daily junk food consumption were both associated with reduced mental and physical health [18]. It is thought that keeping a set interval between meals generates proper feelings of hunger and appetite, which leads to a balanced diet. In addition, it has previously been reported that children who frequently played outside after school and who ate “supplementary foods” instead of “non-essential foods” for snacks consumed more vegetables at dinner [19]. Therefore, it was presumed that feeling hungry before meals is associated with the quantity of vegetables consumed at meals. An irregular diet in preschool children has also been significantly associated with risk factors for lifestyle-related diseases, such as overweight and obesity [20]. Here, more regular mealtimes were associated with higher C-HDS scores, indicating that mealtime regularity is also chronologically associated with dietary balance. Although the results of this study could not demonstrate causal relationships between meal regularity and dietary balance, they do highlight the importance of regular mealtimes. 

In this study, we developed the HDS as an indicator for evaluation of preschool children’s dietary balance. Other indicators of dietary balance include the Health Eating Index (HEI) (2015), which uses the U.S. dietary reference intakes [21] and is calculated as a total score of nine recommended food groups and four items for which intake should be restricted. The HEI has been shown to be associated with child socioeconomic indicators [22] and with the frequency of fast-food consumption and eating dinner together as a family among preschool children [23]. Another such indicator is the diet diversity score (DDS), which evaluates the diversity of meals using various methods. Most use the 24 h memory method, meal records, or food frequency surveys to calculate the intakes of different food groups. The DDS has been reported to be associated with child anthropometric indices [24]. Qualitative dietary indicators evaluate how frequently the different food groups are consumed. Dietary diversity indicators have been reported to be related to BMI in preschool children [25] and parents who consider the content of their children’s meals and snacks and aim to have them eat regular meals [12]. The 2015 National Nutrition Survey on Preschool Children in Japan included simple questions about the frequency of consuming foods from different food groups. Considering the burden these surveys place on respondents, using a similar simple indicator of dietary quality is preferable. 

Japan’s Fourth Basic Plan for the Promotion of Shokuiku set a target for increasing the proportion of the population that eats meals combining staple, main, and side dishes at least twice per day almost daily to more than 50.0%. In addition, the Japanese Food Guide Spinning Top [11] recommends evaluating dietary quality by considering five categories (grain dishes, main dishes, side dishes, fruit, and dairy products). The Mediterranean Diet Quality Index for Children and Adolescents (KIDMED) allocates +1 point for 12 recommended items, such as fruit and vegetables, and −1 point for four items that should be avoided, such as fast foods and sweets. Indicators such as the KIDMED, which also considers unhealthy foods, can also be useful [26]. The usefulness of the HDS and its relationship with other indicators requires further research.

The present study is significant because few studies have examined whether mealtime regularity is related to lifestyle habits and dietary balance in a large group of preschool children, including changes before and after COVID-19. However, this study is subject to some limitations. First, changes in mealtime regularity were categorized into four groups based on responses to a questionnaire. However, a definition of regular was not provided and was left open to interpretation by the respondents. Second, the HDS, which was developed for this study, was calculated according to the intake frequency of 13 food groups, and therefore may be less valid than indicators calculated based on the quantities consumed. Third, because this was an online study of people registered with a survey company, it may have been biased toward people who regularly use the Internet. Fourth, the survey required respondents to recall the situation before the pandemic and to evaluate changes in lifestyle habits and mealtime regularity now compared with before COVID-19. Thus, there is a possibility of recall bias. Fifth, apart from the impact of COVID-19, during the period of a year in early childhood, the content of a child’s diet may also change, owing to growth. Furthermore, a second state of emergency was in effect during February 2021 when this study was being conducted. However, this only applied to certain regions and likely did not cover all the areas where the participants lived. Therefore, owing to differences in the COVID-19 situation throughout the country, the participants may have experienced different restrictions in their daily lives. Furthermore, there may have been individual differences in terms of awareness of and actions taken for infection control.

## 5. Conclusions

Our investigation of how mealtime regularity relates to lifestyle and dietary habits in children indicates that those who originally had regular mealtimes had higher HDS scores and better lifestyle habits, such as waking and going to bed earlier, less snacking, and eating breakfast every day. Furthermore, as C-HDS scores were higher in the group whose mealtimes became regular during the pandemic, adopting regular mealtimes may lead to a more balanced diet. In order to improve children’s dietary balance, it is important to provide nutrition education that not only focuses on the composition of meals but also on mealtime regularity.

## Figures and Tables

**Figure 1 nutrients-14-02979-f001:**
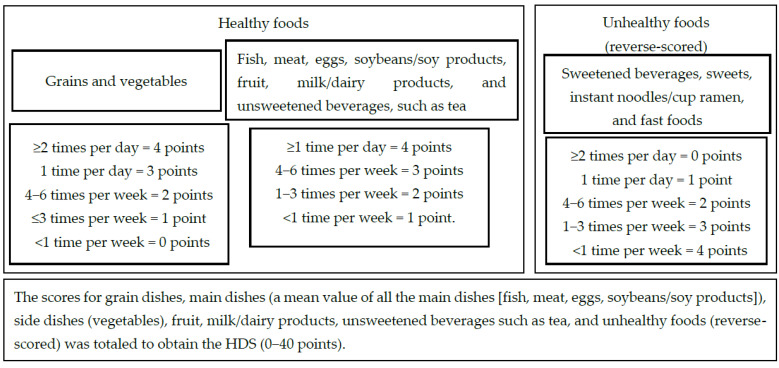
Calculating the healthy diet score (HDS).

**Figure 2 nutrients-14-02979-f002:**
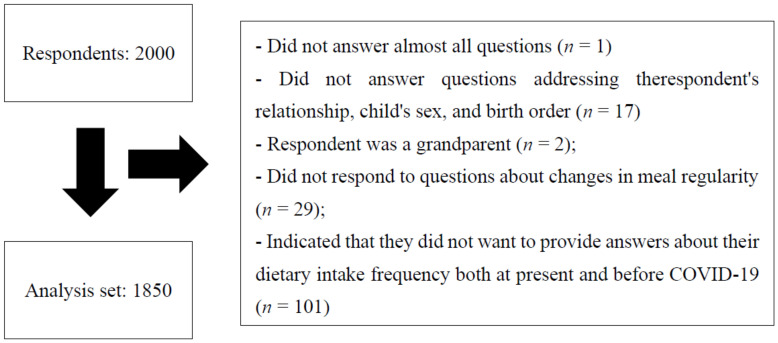
Participant flowchart.

**Table 1 nutrients-14-02979-t001:** Participant characteristics according to changes in mealtime regularity compared with before COVID-19.

		Became Regular		Originally Regular, No Change		Became Irregular		Originally Irregular, No Change	
		(*n* = 125)		(*n* = 1514)		(*n* = 63)		(*n* = 148)	
Respondent’s relationship to the child	Father	56	(44.8)	618	(40.8)	22	(34.9)	74	(50.0)
	Mother	69	(55.2)	896	(59.2)	41	(65.1)	74	(50.0)
Employment status of the child’s mother	Employed	78	(62.4)	827	(54.6)	38	(60.3)	74	(50.0)
	Other	47	(37.6)	687	(45.4)	25	(39.7)	74	(50.0)
Child’s sex	Boy	69	(55.2)	752	(49.7)	30	(47.6)	68	(45.9)
	Girl	56	(44.8)	762	(50.3)	33	(52.4)	80	(54.1)
Child’s age (years)		3.2	±1.4	3.4	±1.4	3.2	±1.4	2.9	±1.4
Height (cm)		105.6	±12.1	105.9	±11.4	105.6	±12.1	102.9	±11.5
Weight (kg)		17.3	±4.6	17.3	±3.9	17.2	±4.1	16.5	±3.7
BMI percentile		46.7	±36.0	44.9	±34.2	39.7	±32.0	49.9	±33.2

Categorical variables are expressed as number of people (percentage), and continuous variables are expressed as mean ± standard deviation. BMI, body mass index.

**Table 2 nutrients-14-02979-t002:** Lifestyle habits and household circumstances according to changes in mealtime regularity compared with before COVID-19.

		Became Regular		Originally Regular, No Change		Became Irregular		Originally Irregular, No Change		*p* Value *
		(*n* = 125)		(*n* = 1514)		(*n* = 63)		(*n* = 148)		
Wakeup time	Before 8 a.m. on weekdays and holidays	83	(66.4)	**1066**	**(70.4)**	32	(50.8)	54	(36.5)	<0.001
	Other	42	(33.6)	448	(29.6)	**31**	**(49.2)**	**94**	**(63.5)**	
Bed time	Before 10 p.m. on weekdays and holidays	91	(72.8)	**1156**	**(76.4)**	34	(54.0)	53	(35.8)	<0.001
	Other	34	(27.2)	358	(23.6)	**29**	**(46.0)**	**95**	**(64.2)**	
Physical activity time (weekdays)	<1 h (/day)	68	(54.4)	708	(47.0)	35	(55.6)	74	(51.0)	0.200
	≥1 h (/day)	57	(45.6)	**799**	**(53.0)**	28	(44.4)	71	(49.0)	
Physical activity time (holidays)	<1 h (/day)	69	(55.2)	743	(49.3)	35	(55.6)	74	(50.7)	0.485
	≥1 h (/day)	56	(44.8)	764	(50.7)	28	(44.4)	72	(49.3)	
Screen time (weekdays)	<2 h (/day)	97	(79.5)	**1219**	**(80.8)**	44	(71.0)	98	(66.7)	<0.001
	≥2 h (/day)	25	(20.5)	289	(19.2)	18	(29.0)	**49**	**(33.3)**	
Screen time (holidays)	<2 h (/day)	85	(69.7)	983	(65.2)	37	(59.7)	76	(52.4)	0.009
	≥2 h (/day)	37	(30.3)	525	(34.8)	25	(40.3)	**69**	**(47.6)**	
Frequency of defecation	Almost every day	94	(76.4)	1072	(70.9)	34	(54.0)	105	(71.4)	0.014
	Other	29	(23.6)	439	(29.1)	**29**	**(46.0)**	42	(28.6)	
Snack frequency	0−1 time (/day)	79	(64.8)	**1056**	**(70.1)**	31	(50.0)	84	(57.9)	<0.001
	2 times (/day)	37	(30.3)	390	(25.9)	**25**	**(40.3)**	40	(27.6)	
	≥3 times (/day)	6	(4.9)	60	(4.0)	6	(9.7)	**21**	**(14.5)**	
Breakfast frequency	Eat every day	112	(89.6)	**1442**	**(95.2)**	47	(74.6)	116	(78.4)	<0.001
	Other	13	(10.4)	72	(4.8)	**16**	**(25.4)**	**32**	**(21.6)**	
Eat together (breakfast)	Eat with an adult	106	(86.2)	**1249**	**(82.7)**	51	(81.0)	97	(67.4)	<0.001
	Other	17	(13.8)	262	(17.3)	12	(19.0)	**47**	**(32.6)**	
Eat together (dinner)	Eat with an adult	122	(97.6)	1457	(96.4)	61	(96.8)	135	(91.8)	0.041
	Other	3	(2.4)	55	(3.6)	2	(3.2)	**12**	**(8.2)**	
How often the respondent cooks	1−2 times per week	36	(28.8)	469	(31.2)	20	(31.7)	55	(37.9)	0.474
	3−6 times per week	27	(21.6)	330	(22.0)	17	(27.0)	34	(23.4)	
	Every day	62	(49.6)	702	(46.8)	26	(41.3)	56	(38.6)	
Frequency with which the child is involved in cooking	Rarely	41	(32.8)	672	(44.4)	17	(27.4)	**82**	**(55.4)**	<0.001
	≥1 time per week	**84**	**(67.2)**	840	(55.6)	**45**	**(72.6)**	66	(44.6)	
Someone in the family works remotely		**51**	**(40.8)**	390	(25.8)	**25**	**(39.7)**	28	(18.9)	<0.001
Financial security	Yes	54	(43.5)	567	(37.5)	24	(38.1)	38	(25.7)	0.003
	Neither yes nor no	42	(33.9)	459	(30.4)	19	(30.2)	42	(28.4)	
	No	28	(22.6)	484	(32.1)	20	(31.7)	**68**	**(45.9)**	
Has free time	Yes	40	(32.3)	452	(29.9)	16	(25.4)	29	(19.6)	0.039
	Neither yes nor no	44	(35.5)	479	(31.7)	15	(23.8)	52	(35.1)	
	No	40	(32.3)	580	(38.4)	32	(50.8)	67	(45.3)	

The continuous variables presented in the table are expressed as mean ± standard deviation. * Chi-squared test (residual analysis performed when significant differences were noted; underlined numbers indicate adjusted standardized residual <−1.96; bold numbers indicate adjusted standardized residual >1.96).

**Table 3 nutrients-14-02979-t003:** Comparison of healthy diet score (HDS) according to changes in mealtime regularity compared with before COVID-19.

	Became Regular			Originally Regular, No Change			Became Irregular			Originally Irregular, No Change			*p* Value
	(*n* = 125)			(*n* = 1516)			(*n* = 63)			(*n* = 148)			
Current intake of the different food groups													
**Healthy Diet Score * (0−40 points)**	**29.7**	**±4.7**	** ^a^ **	**31.6**	**±4.0**	** ^abc^ **	**29.6**	**±4.8**	** ^b^ **	**28.3**	**±4.6**	** ^c^ **	**<0.001**
Grain dishes, main dishes (mean), and side dishes (0−12 points)	9.1	±2.4	^a^	9.3	±1.8	^b^	8.6	±2.5		8.2	±2.2	^ab^	<0.001
Grain dishes, main dishes (mean), side dishes, fruit, and dairy products (0−20 points)	15.3	±3.8	^a^	15.7	±3.0	^b^	14.8	±3.9		13.9	±3.3	^ab^	<0.001
Grain dishes, main dishes (total), side dishes, fruit, and dairy products ** (0−32 points)	23.8	±6.0	^a^	23.8	±4.6	^b^	22.8	±5.7		21.1	±5.1	^ab^	<0.001
Infrequently consuming unhealthy foods (0−16 points) ^†^	11.0	±3.8	^a^	12.2	±2.7	^ab^	11.4	±3.3		11.0	±2.9	^b^	<0.001
Changes compared with before the pandemic ^#^													
**Changed Healthy Diet Score ^#^ (−13~+13)**	**0.8**	**±2.3**	** ^abc^ **	**0.1**	**±1.3**	** ^a^ **	**−0.1**	**±1.6**	** ^b^ **	**0.0**	**±1.0**	** ^c^ **	**<0.001**
Increased score (−4~9 points)	0.9	±2.0	^ab^	0.1	±1.1	^a^	0.3	±1.5		0.0	±0.9	^b^	<0.001
Decreased score (−9~4 points)	−0.1	±1.0		0.0	±0.5	^a^	−0.4	±0.9	^ab^	0.0	±0.5	^b^	<0.001

The continuous variables presented in the table are expressed as mean ± standard deviation. One-way analysis of variance (multiple comparisons using the Games–Howell method; ^a–c^, performed for significant differences). HDS, healthy diet score; C-HDS, changed healthy diet score. * The intake frequency of each food group was determined and used to calculate the total intake frequency of grain dishes; main dishes (mean); side dishes; fruit; dairy products; unsweetened foods, such as tea; and unhealthy foods (reverse-scored question) as follows: grains and vegetables: ≥2 times per day = 4 points, 1 time per day = 3 points, 4−6 times per week = 2 points, ≤3 times per week = 1 point, <1 time per week = 0 points; fish, meat, eggs, soybeans/soy products, fruit, milk/dairy products, and unsweetened beverages such as tea: ≥1 time per day = 4 points, 4−6 times per week = 3 points, 1−3 times per week = 2 points, <1 time per week = 1 point. The intake frequency of unhealthy foods, such as sweetened beverages, sweets, instant noodles/cup ramen, and fast foods, was reverse-scored as follows: ≥2 times per day = 0 points, 1 time per day = 1 point, 4−6 times per week = 2 points, 1−3 times per week = 3 points, and <1 time per week = 4 points. ** Indicator considering the diversity of main dishes by using the total of the main dishes (fish, meat, eggs, and soybeans/soy products). ^†^ Healthy foods were defined as grains; fish; meat; eggs; soybeans/soy products; vegetables; fruit; milk/dairy products; and unsweetened beverages, such as tea. Unhealthy foods were defined as sweetened beverages, sweets, instant noodles/cup ramen, and fast foods. ^#^ The change in intake frequency for a food group before and after COVID-19 was scored as follows: for increased intakes of a specific food group, healthy foods = +1 point, unhealthy foods = −1 point, and no change = 0 points (increased score −4 to 9 points); and for decreased intakes of a specific food group, healthy foods = −1 point, unhealthy foods = +1 point, and no change = 0 points (decreased score: −9 to 4 points). These were totaled to determine the C-HDS.

**Table 4 nutrients-14-02979-t004:** Multiple regression analyses with the healthy diet score (HDS) as the dependent variable.

Item Category		Simple Regression		Model 1		Model 2		
		Standardized Coefficient β	*p* Value	Standardized Coefficient β	*p* Value	Standardized Coefficient	*p* Value	
Basic attributes	Relationship to child	0.163	<0.001	0.156	<0.001	0.110	<0.001	^#^
	Employment status of child’s mother	−0.061	0.009	−0.038	0.119	−0.006	0.788	
	Child’s sex	0.018	0.448	0.026	0.280	0.040	0.070	
	Child’s age	−0.043	0.065	−0.019	0.425	−0.075	0.001	^#^
	BMI percentile	−0.051	0.037	−0.036	0.137	−0.016	0.462	
Lifestyle habits	Wakeup time (weekdays and holidays)	−0.143	<0.001	−0.052	0.028	−0.054	0.025	^#^
	Bed time (weekdays and holidays)	−0.174	<0.001	−0.079	0.001	−0.093	<0.001	^#^
	Physical activity time (weekdays)	0.079	0.008	0.033	0.196			
	Physical activity time (holidays)	0.079	0.001	0.045	0.075	0.077	<0.001	^#^
	Screen time (weekdays)	−0.101	<0.001	−0.009	0.757			
	Screen time (holidays)	−0.089	0.001	−0.027	0.326			
	Frequency of defecation	0.136	<0.001	0.101	<0.001	0.107	<0.001	^#^
	Snack frequency	−0.291	<0.001	−0.233	<0.001	−0.236	<0.001	^#^
	Breakfast frequency	0.249	<0.001	0.141	<0.001	0.137	<0.001	^#^
	**Originally regular meal times**	**0.229**	**<0.001**	**0.139**	**<0.001**	**0.131**	**<0.001**	^#^
Household circumstances	Eat together (breakfast)	0.070	0.003	0.059	0.013	0.022	0.326	
	Eat together (dinner)	0.062	0.008	0.051	0.031	0.034	0.134	
	How often the respondent cooks							
	Every day	0.192	<0.001	0.187	<0.001	0.092	0.007	^#^
	3−6 times a week	−0.099	<0.001	−0.007	0.795	−0.013	0.650	
	Rarely	-		-		-		
	Frequency with which the child is involved in cooking	0.034	0.149	0.017	0.468			
	Someone in the family works remotely	−0.024	0.305	−0.022	0.336			
	Financial security							
	Yes	0.067	0.004	0.028	0.319	0.021	0.423	
	No	−0.062	0.008	−0.061	0.034	−0.021	0.419	
	Neither yes nor no	-		-		-		
	Has free time							
	Yes	0.033	0.151	0.035	0.202			
	No	0.011	0.647	0.052	0.064			
	Neither yes nor no	-		-		-		

Model 1, multiple regression analysis with forced entry of all items for each category. Model 2, multiple regression analysis with forced entry of the basic attributes and items that exhibited significant associations with the HDS in model 1. ^#^
*p* < 0.05 in the multiple regression analysis using data after multiple imputation of missing values. BMI, body mass index. Input variables: respondent’s relationship to the child (0, father; 1, mother); mother’s employment status (1, working; 0, other); child’s sex (0, boy; 1, girl); child’s age (continuous variable); BMI percentile (0, <75th percentile; 1, ≥75th percentile); wakeup time (0, weekdays and holidays before 8 a.m.; 1, otherwise); bed time (0, weekdays and holidays before 10 p.m.; 1, otherwise); physical activity time (weekdays and holidays) (0, <1 h; 1, ≥1 h); screen time (weekdays and holidays) (0, <2 h; 1, ≥2 h); defecation frequency (1, daily; 0, other); snack frequency (1, 0−1 time/day; 2, 2 times/day; 3, ≥3 times/day); frequency of breakfast (1, daily; 0, other); eating together (1, eat with an adult; 0, other); how often the respondent cooks (“rarely” input as reference value); frequency with which the child is involved in cooking (0, rarely; 1, ≥1 day/week); someone in household works remotely (1, yes; 0, no); and financial security/has free time (“not a lot” input as reference value).

**Table 5 nutrients-14-02979-t005:** Multiple logistic regression analysis with the change in healthy diet score (C-HDS) as the dependent variable.

		*n* (%)		Increase ^#^				*p* Value	Decrease ^#^				*p* Value
				OR (95% CI)					OR (95% CI)				
Meal regularity	**Became regular**	**125**	**(6.7)**	**2.21**	**(1.35**	**−**	**3.61)**	**0.002**	1.50	(0.80	−	2.80)	0.205
	Became irregular	63	(3.4)	1.23	(0.60	−	2.53)	0.564	1.85	(0.92	−	3.72)	0.083
	No change	1664	(89.8)	1.00					1.00				

Multiple logistic regression analysis using data after multiple imputation of missing values (*n* = 1825). ^#^ Dependent variable reference: no change (C-HDS = 0 points). BMI, body mass index; CI, confidence interval; OR, odds ratio. Input variables: respondent’s relationship to the child (0, father; 1, mother); mother’s employment status (1, working; 0, other); child’s sex (1, boy; 2, girl); child’s age (continuous variable); BMI percentile (0, <75th percentile; 1, ≥75th percentile); change in wakeup and bed time (became regular, became irregular, or unchanged); change in the frequency and time of physical activity (increased, decreased, or unchanged); snack frequency (increased, decreased, or unchanged); breakfast frequency (increased, decreased, or unchanged); and how often the respondent cooks (increased, decreased, or unchanged).

## Data Availability

The data sets used and/or analyzed during the current study are available from the corresponding author on reasonable request.

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
