# Peer review of "Mealtime Regularity Is Associated with Dietary Balance among Preschool Children in Japan—A Study of Lifestyle Changes during the COVID-19 Pandemic"

_nutrients, 2022, doi:10.3390/nu14142979_

Round 1
Reviewer 1 Report
The paper submitted for review is related to the topic of lifestyle changes during the COVID-19 pandemic. In their study, the authors addressed the aspect of meal regularity in a group of preschool children. The overall reception of the manuscript is very good. The paper seems to be well organized and meets all the formal requirements for papers published in MDPI journals.
Nevertheless, I have a few suggestions that would definitely enrich the work visually.
- please verify the citations in the text, sometimes the double cross-reference is written as [X,Y], sometimes [X][Y], which is completely incomprehensible by me.
- please do not quote the same source in a row. It is completely sufficient to add a cross-reference at the end of the quoted thought - example: line 133.
- the purpose of the study is understandable, but isn't it worth expanding it to include research questions? - I leave it to the Authors to decide.
- did the Authors use any methods/systems to prevent bot/fake responders in conducting the online survey?
- the description of HDS in section 2.2. would look better in tabular form.
- Please write the p-value in lowercase - example: line 161.
- in the tables, I recommend bolding the more important results to make it easier to read. You can also use the color method and supplement it with a legend under the table.
- designations a, b, c, in tables 2 and 3 may not be understandable to the reader. Please explain them in detail or use another method to mark the importance of results.
- part of the strengths and weaknesses should be extracted from the discussion. Why don't the authors give the advantages of their study?
- the conclusions should be more widely described, because they do not reflect the amount of work put in by the authors.
Kind regards!
Author Response
To reviewer #1:
Thank you very much for reviewing our manuscript. We reconsidered and revised the manuscript to address your suggestions. We appreciate your review of this work.
Comment:
- please verify the citations in the text, sometimes the double cross-reference is written as [X,Y], sometimes [X][Y], which is completely incomprehensible by me.
Answer:
Thank you for pointing this out. We have corrected the overlapping parentheses to a comma-separated expression (LINE 134), but ultimately removed it upon your next comment.
Comment:
- please do not quote the same source in a row. It is completely sufficient to add a cross-reference at the end of the quoted thought - example: line 133.
Answer:
The two papers cited in line 133 have been removed here because they are cited again in the Discussion and develop similar content.
Comment:
- the purpose of the study is understandable, but isn't it worth expanding it to include research questions? - I leave it to the Authors to decide.
Answer:
Thank you for your suggestion. We consider the last sentence of the previous paragraph to be a research question.
LINE 98;
In particular, it was unclear whether the dietary habits of children who maintained their mealtime regularity during the pandemic were more balanced than those who did not.
Comment:
- did the Authors use any methods/systems to prevent bot/fake responders in conducting the online survey?
Answer:
In order to eliminate spoofed panels and obtain correct information, Company C conducts regular attribute checks. Furthermore, we excluded the participant from the analysis who did not answer almost all the questions and who did not provide answers about their relationship to the child, the child's sex, or birth order to prevent impersonation and fraudulent responses. Furthermore, inconsistent answers for individual items were excluded from the final analysis. For example, those who indicated that their children are now “rarely” involved in cooking but “considerably more” than before the COVID-19 (n=2) were considered to be inconsistent answers. Participants who reported eating breakfast with their family members more often than before COVID-19, even though they currently reported eating breakfast [with children alone | alone] (n=7), participants who reported eating dinner with their family members more often than before COVID-19, even though they currently reported eating dinner [with children alone | alone] (n=2), participants who reported an [increase] in the amount of time physically active on weekdays/holidays compared to before COVID-19 (1 weekday, 1 holiday), even though they reported that they currently [do not do this at all], and participants who reported an [increase] in the amount of time their child watches TV or videos at home on weekdays/holidays compared to before COVID-19 (11 weekdays, 10 holidays), even though they reported that they currently [do not watch or do this], were also excluded. Statements regarding height and weight, values over the 99th percentile or below the 1st percentile (height, n=71 participants and weight, n=33 participants) are also kept in mind to exclude fraudulent responses. In the section describing the subject's flowchart, the description was modified to convey this as below. Since describing all the excluded contradictory responses would be too long to explain, only some of them are exemplified.
LINE 205;
To prevent impersonation and fraudulent responses, 1 participant who did not answer almost all the questions and 17 participants who did not provide answers about their relationship to the child, the child's sex, or birth order were excluded from the analysis. Furthermore, 2 participants who were grandparents; 29 participants who did not provide answers for questions addressing changes in meal regularity; and 101 participants who indicated that they did not want to provide answers about their dietary intake frequency at present or before/after COVID-19 were excluded from the analysis; finally, 1850 participants were included in the final analysis (Figure 2).
Inconsistent answers for individual items were excluded from the final analysis. For example, answers that indicated that the respondent’s children are now “rarely” involved in cooking but “considerably more” than before the COVID-19 (n=2) were considered to be inconsistent answers. Regarding a child’s height and weight, values over the 99th percentile or below the 1st percentile and calculated using the least squares mean method were considered input errors and excluded from the calculation of obesity (height, n=71 participants and weight, n=33 participants).
Comment:
- the description of HDS in section 2.2. would look better in tabular form.
Answer:
We agree with your comment. We have organized our calculations in the form of a diagram (figure 1).
Comment:
- Please write the p-value in lowercase - example: line 161.
Answer:
We have corrected the p-values in the text to lowercase.
Comment:
- in the tables, I recommend bolding the more important results to make it easier to read. You can also use the color method and supplement it with a legend under the table.
Answer:
We have bolded the main HDS score (40-points) and Changed Healthy Diet Score (-13−+13) to make them stand out from the subscales in Table 3. We have also bolded to highlight the dietary regularity results in Table 4 (originally regular meal times) and Table 5 (became regular).
Comment:
- designations a, b, c, in tables 2 and 3 may not be understandable to the reader. Please explain them in detail or use another method to mark the importance of results.
Answer:
We have added details of the residual analysis to the footnotes in Table 2. We have also changed the symbols in Tables 2, as we considered the fact that the same a−c symbols indicated different meanings in Tables 2 and 3 to be a confusing factor.
The footnote in table 2;
*The chi-squared test (residual analysis performed when significant differences were noted; underlined numbers indicate adjusted standardized residual <–1.96; bold numbers indicate adjusted standardized residual >1.96).
Comment:
- part of the strengths and weaknesses should be extracted from the discussion. Why don't the authors give the advantages of their study?
Answer:
We have added a statement about the strengths at the beginning of the Limitations section, as shown below.
LINE 453;
The present study is significant because few studies have examined whether the mealtime regularity is related to lifestyle habits and dietary balance in a large group of preschool children, including changes before and after COVID-19. However, this study had some limitations.
Comment:
- the conclusions should be more widely described, because they do not reflect the amount of work put in by the authors.
Answer:
We agree with your comment. We have expanded the conclusions as shown below.
Line 474;
Our investigation of how mealtime regularity relates to lifestyle and dietary habits in children indicated that those who originally had regular mealtimes had higher HDS and better lifestyle habits, such as waking and going to bed earlier, less snacking, and eating breakfast every day. Further, as the C-HDS was higher in the group whose mealtimes became regular during the pandemic, adopting regular mealtimes may lead to a more balanced diet. In order to improve children's dietary balance, it is important to provide nutrition education that not only focuses on the composition of meals, but also on the mealtime regularity.

Reviewer 2 Report
The present study makes a comprehensive analysis of the effects of Mealtime regularity is associated with dietary balance among preschool children in Japan, based on the lifestyle changes produced by COVID-19.
Overall the experimental design is adequate, the sample size is sufficient for robust results, and the analysis is complete.
The authors have analyzed all the variables involved in the study (nutritional, social, and lifestyle variables) and the presentation of the data is clear, with a complete and adequate analysis of the results, mostly based on updated references.
This type of data is necessary for the scientific community, as it helps us to prepare for future pandemics.
There are only a few observations to correct:
Correct the wording of the abstract, there are many repeated words.
It is convenient that the comparison with similar studies is in the result analysis section and not in the introduction.
Limit the introduction to the approach and context of the problem, define some concepts if necessary and focus on the main objective.
All the information is in the tables, but in the results paragraphs, it is necessary to highlight the important results observed in the tables.
Check the English grammar throughout the document.
Author Response
To reviewer #2:
Thank you very much for reviewing our manuscript. We have reconsidered and revised the manuscript to address your suggestions. We appreciate your review of this work.
Comment:
Correct the wording of the abstract, there are many repeated words.
Answer:
We agree with your comment. We have revised the text to eliminate duplication and express our intentions clearly, as shown below.
LINE 25;
This study aimed to determine whether the pandemic affected mealtime regularity among preschool children, and whether maintaining regular mealtimes or changes in mealtime regularity during the pandemic were related to dietary balance, including chronological relationships.
Comment:
It is convenient that the comparison with similar studies is in the result analysis section and not in the introduction.
Answer:
We agree with your comment. Several references were removed from the Introduction section and reorganized.
Comment:
Limit the introduction to the approach and context of the problem, define some concepts if necessary and focus on the main objective.
Answer:
As you suggested, we have simplified the Introduction section to focus on the main objective of the study. In particular, the Assistant Editor pointed out that there were too many references from the same journal, namely Nutrients; hence, we have deleted two sentences with references from Nutrients.
Comment:
All the information is in the tables, but in the results paragraphs, it is necessary to highlight the important results observed in the tables.
Answer:
We agree with your comment. The results that we consider particularly important are in bold in the tables and have been further explained in the text.
Comment:
Check the English grammar throughout the document.
Answer:
Thank you for pointing this out. We have had the revised manuscript proofread by a professional English editing service prior to resubmission.
